# Factors Influencing Students’ Career Intentions in the Hospitality and Tourism Industries: A Meta-Analysis

**DOI:** 10.3390/bs12120517

**Published:** 2022-12-16

**Authors:** Fajian Liu, Qing He, Nan Wu

**Affiliations:** 1Huizhou Studies Research Center, Anhui University, Hefei 230039, China; 2School of Business, Anhui University, Hefei 230601, China

**Keywords:** career intentions, influencing factors, hospitality and tourism management students, meta-analysis

## Abstract

This study adopted a meta-analysis to generalize and compare the effect of influencing factors on hospitality and tourism management students’ career intentions from 34 studies. After 13 existing factors were classified into four categories, the results revealed that 11 of them significantly impact students’ career intentions. In general, the influences of social identity and self-perception are more important, followed by school education and work conditions. A subgroup analysis further identified the sample students’ grades as dynamics that partially moderate some of the influencing factors, while no statistical changes in the relative influences were observed after the COVID-19 outbreak. Finally, suggestions are provided for hospitality and tourism management education and industries.

## 1. Introduction

With the rapid growth of the hospitality and tourism (H&T) industries worldwide, there has been an increasing demand for tourism practitioners in various countries, which has led to the rapid development of tourism education [1]. There is no doubt that education plays a pivotal role in providing well-trained, qualified staff with a wide range of expertise, especially in the H&T industry [2]. A key issue facing the H&T industry is how to position itself as an ideal workplace for young people [3]. The problem of the H&T industry’s employment prospects has not been notably improved, and hospitality and tourism management (HTM) students’ willingness to stay in the industry is low; thus, H&T industries worldwide are facing distinct challenges in attracting and retaining talent [4,5]. Notably, it has been getting worse since the outbreak of COVID-19. The answer lies not only in escalating uncertainties and negative emotions about the H&T industries caused by the pandemic [6,7], but also in the mismatch of supply and demand between the industry and potential students [8]. Accordingly, identifying the factors that lead to low career intentions in the H&T industry and the high turnover rate of graduates that this has caused is a prerequisite for solving the dilemma of the talent shortage in the H&T industry [9]. It has become a concern for industries, destination marketing organizations (DMOs), and HTM education providers.

Currently, many studies have tried to determine what factors affect students’ career intentions (CIs), willingness, and motivations, as well as how to improve them [10]. Student personality traits [11,12], curriculum and internship arrangements [3,13,14], job-related factors [5,15,16,17], parental support and help [18], the COVID-19 pandemic [6,7,19], and many other factors have a demonstrated relationship with students’ CIs. From these studies, at different times, it can be seen that many factors (e.g., salary, career prospects) have always been key factors affecting HTM students’ employment. However, differences and even opposing results exist about the relationship direction and strength of those same factors to students’ intentions, which might be caused by differences in samples, conditions, and other contexts. For example, while some studies have suggested the positive effects of salary on HTM students’ CIs [20], others found that this effect was negligible [5,14]. Chuang et al. [21] and Wang [22] proved the positive influence of outcome expectations on their students’ CIs, but a negative relationship was also obtained [23]. Indeed, a consensus regarding the strength and direction of the relationships cannot easily be reached, while a systematic overview of the factors affecting students’ intentions is lacking, as previous studies mostly focused on one aspect of them and their results are related to their specific context. Meanwhile, there are still specific doubts about the more important factors. Therefore, we should synchronize the overall effect of these factors into an integrated framework in order to eliminate the influence of research methods, cases, and backgrounds.

This study aims to quantitatively draw comprehensive conclusions about the relationships between HTM students’ CIs and various influencing factors; a meta-analysis is employed for this purpose. By synthesizing the results of relevant studies, the meta-analysis systematically verifies the overall effect of the factors and eliminates the biases of and disagreements in existing studies [24]. Here, we not only test and compare the influences of various factors on students’ CIs, but also assess the differences in research contexts, including the sample’s grades and study time periods. It is hoped that the results can provide effective suggestions for universities, industries, and DMOs to amplify positive factors and overcome negatives to influence students’ perceptions of and satisfaction with the H&T industries and further enhance their CIs.

## 2. Literature Review and Conceptual Framework

### 2.1. CIs and Related Influencing Factors

For HTM students, their CIs within the industry refer to the extent of their behavioral intentions to continue to choose or accept a job within the H&T industries as their official job after graduation [25]. Normally, some expressions, such as willingness, commitment, aspiration, engagement, and choice, represent positive attitudes in different studies, while one’s career can also be expressed as one’s employment or profession [4,10,21,26]. Here, we unified these expressions into the term “career intention” in this paper, and the literature about the above variables was also collected into the meta-analysis. According to the theory of planned behavior, intention is considered a valid predictor of future behavior [27]. Previous studies have shown that students’ CIs are closely related to their perceptions and attitudes toward work [28], and they are very important to increase the possibility of HTM students pursuing work [15,16].

The formation of CIs or attitudes is a complex, multifactorial process, and many studies have concentrated on the relevant influencing factors. Teng [11] confirmed the influence of students’ personality traits and attitudes on their pursuit of hospitality jobs. Wang [23] discussed the impacts of tourism students’ motivation to complete their majors, the effects of their learning, and their perceptions of H&T industry prospects on their career choices. In addition, many studies have specifically examined the impact of internships on HTM students’ CIs [29,30]. By getting in touch with real jobs through internships, students formed different feelings about the H&T industries, such as the potential job satisfaction these industries provide and their perception of the interpersonal relationships within them, which also impact their willingness to enter these industries after graduation. Lee and Chao [20] discussed internship organization factors that affect the employment of hospitality students. In fact, many of the influencing factors in HTM students’ CIs appeared to be obstacles. Kusluvan and Kusluvan [15] found that they generally hold a negative attitude toward working in the tourism industry because of its high work pressure, low benefits, low social status, and unfair promotion practices. Similarly, Wu [31] found that social status, family responsibility and pressure, and education were obstructive factors.

Moreover, numerous factors from these studies can also be generalized into some categories. Jiang and Tribe [32] constructed five subcategory factors that affect students’ engagement in the tourism industry, including personal, educational, and managerial factors, etc. Lee et al. [33] identified five categories of factors influencing hospitality management students’ choices, namely, faculty members, advisors, parents, etc. Gong and Jia [10] classified factors into external and internal types. These categories provide an essential reference and theoretical background to draw upon when creating a framework of influencing factors. Indeed, some previous studies focused on a specific category of factors, even a single factor, and conflicting conclusions were often obtained from differing contexts. On the other hand, multifactor studies weighted toward qualitative classification lacked quantitative measurement and comparison. We attempted to summarize an integrated framework that categorizes the various factors and to then conduct quantitative research to eliminate their differences.

### 2.2. Study Framework and Hypotheses

Considering the requirements of meta-analysis methods and existing studies, we teased out 13 influencing factors that have been frequently quantitatively studied in the literature, and classified them into four categories: work conditions, social identity, school education, and self-perception. Furthermore, we also considered sample characteristics and the time period as the research setting, which also have an inevitable effect on the results. The framework of influencing factors of HTM students’ CIs is as follows (see Figure 1).

#### 2.2.1. CIs and Work Condition Factors

Students’ CIs are often affected by some conditions the job provides. They could be generalized as the material, psychological, and physical environment. Four specific factors are focused on here: the nature of the work, salary, environmental conditions, and interpersonal relationships.

Firstly, the nature of the work in the H&T industry is considered a critical factor that affects students’ career commitment [15,22]. It reflects the characteristics of the work itself, such as the work content, the skills required, and the working hours. These objective conditions can trigger certain subjective feelings in students or employees. For example, they can be affected by whether it is a challenging job and aligns with their interests [11,15]. Meanwhile, economic factors are the main influencing factors for undergraduates’ choice of career direction, including salary and benefits; the same might be valid for HTM students [34]. Chuang and Dellmann-Jenkins [21] found that students showed a greater willingness to be employed in the hospitality industry if the industry offered rewards and incentives as expected. A similar conclusion was obtained by Lee and Chao’s [20] study. In addition, comfortable work environmental conditions, such as employee bathrooms, dining halls, and dormitories, as well as the quietness and cleanliness of the working environment, are critical in attracting graduates to engage in the H&T industries. Richardson [16] found that it was very important for HTM students to have a job with a pleasant work environment. However, low pay and poor work environments are prevalent in the H&T industries, which often prevent students from entering the industry after graduation [15].

Moreover, interpersonal relationships in the workplace fall under the category of mental conditions. Interpersonal relationships in the organization will impact students’ job satisfaction, which could also influence their CIs [14,35]. Lee and Chao [20] suggested that when hospitality students receive interpersonal recognition during internships, they feel valued, thereby reducing attrition rates. Therefore, the following hypotheses are proposed:

**Hypotheses** **1.***The nature of the work has a significant impact on HTM students’ CIs*.

**Hypotheses** **2.***Salary has a significant impact on HTM students’ CIs*.

**Hypotheses** **3.***The environmental conditions have a significant impact on HTM students’ CIs*.

**Hypotheses** **4.***Interpersonal relationships have a significant impact on HTM students’ CIs*.

#### 2.2.2. CIs and Social Identity Factors

Identity refers to the active and dynamic understanding of oneself and is obtained from the interaction between the self and the environment [36]. In this study, the social identity of careers is considered to be the students’ perception of the attitudes of their social relations, such as their relatives and friends, toward the occupation they pursue. Among them, social status and career prospects are the main aspects of societal concern [15]. Jiang and Tribe [32] pointed out the high turnover rate among tourism students was partly due to the jobs’ low social status. Tan et al. [17] also found a strong correlation between social status and students’ commitment to pursuing H&T careers, and social status proved to be the main barrier to CIs.

Career prospects represent the individual’s and their relatives’ opinions about the future development and success of their career. Unguren and Huseyinli [1] pointed out that society’s negative views of limited career opportunities hindered student employment in the tourism industry. Wang [37] suggested that the more optimistically graduates perceive their career prospects in the tourism industry, the stronger their willingness is to join it. A similar finding was obtained in a study of Hong Kong HTM students [5]. Accordingly, the study proposed the following hypotheses:

**Hypotheses** **5.***Social status has a significant impact on HTM students’ CIs*.

**Hypotheses** **6.***Career prospect has a significant impact on HTM students’ CIs*.

#### 2.2.3. CIs and School Education Factors

School education factors concerning students’ employment issues mainly include curriculum arrangements, internships, employment guidance, teaching, learning environment, etc. [22,32,33,38]. The actual situation of HTM education is not satisfying. Many problems in the cultivation of tourism professionals still exist, which are the leading causes of the mismatch between the supply and demand of talent in the tourism industry [39]. Here we focused on two factors: education quality and internships; the former reflects the overall situation of the curriculum arrangements, educators’ teaching methodology, and learning environment, while the later represents opportunities to gain access to H&T jobs provided by the educational institution.

Kahraman and Alrawadieh [40] pointed that when tourism students perceived their education was of a high quality, they were more likely to join the industry after graduation. Specifically, Chang and Tse [3] found that graduates working in the H&T industries paid more attention to the quality of HTM foundation courses; the main elective courses were also closely related to their career development. Wang [37] suggested a significant correlation between course satisfaction and hospitality students’ willingness to be employed. Educators are also critical to students’ employment decisions [33]. 

Notably, as a valuable component in HTM education [4,29], internships combine classroom learning with real-world experience and provide an efficient way for students to apply, validate, and integrate what is learned in school. In the process, students gain practical working experience and prepare for work after graduation [41]; hospitality undergraduates regard job-related experiences as the main reference for their career decisions [16]. Therefore, a reasonable and satisfactory internship arrangement, which includes a proper setting for the internship’s job content, targeted position training, and efficient interventions and guidance, can improve students’ attitude toward H&T jobs, and thus increase their intentions to engage in the H&T industries [14,42]. In contrast, unpleasant internship experiences will fail to meet students’ expectations [9], leading them to form negative attitudes and further hindering their entry into the industries [43]. We thus hypothesized the following:

**Hypotheses** **7.***Education quality has a significant impact on HTM students’ CIs*.

**Hypotheses** **8.***Internships have a significant impact on HTM students’ CIs*.

#### 2.2.4. CIs and Self-Perception Factors

Self-perception factors in this study indicate the impact factors related to students’ cognition, evaluation, and preference for work in the industry. Here, we collect data on employability, self-efficacy, person–organization fit, job satisfaction, and outcome expectations, which have been tested in previous literature [44,45].

Employability is an essential quality for graduates to obtain and maintain employment [46]. Wang and Tsai [47] concluded that the more employability students possessed and perceived, the more likely they were to overcome anxiety in obtaining a job and achieve career success after graduation. Self-efficacy reflects an individual’s belief in their ability to complete a certain behavior; it is also closely associated with career planning and decision making in hospitality undergraduates [41]. Person–organization fit refers to one’s subjective assessment of compatibility with the employing organization [48]; it was found to be a significant positive predictor of students’ intentions to work in hotels [11]. Walsh et al. [45] indicated students choose a job that fits their personality characteristics. High job satisfaction leads to better performance, higher involvement, and lower turnover intentions [49]. Dickerson [50] found that most individuals left the hospitality industry because of dissatisfaction with their first hospitality jobs. Lastly, outcome expectation is an important determinant of career interests and choice goals [51]. Many studies have confirmed that students’ willingness to enter the industry is influenced by their career outcome expectations [21,22]. The discussions of these findings led to the following hypotheses:

**Hypotheses** **9.**
*Employability has a significant impact on HTM students’ CIs.*


**Hypotheses** **10.***Self-efficacy has a significant impact on HTM students’ CIs*.

**Hypotheses** **11.***Person–organization fit has a significant impact on HTM students’ CIs*.

**Hypotheses** **12.***Job satisfaction has a significant impact on HTM students’ CIs*.

**Hypotheses** **13.***Outcome expectation has a significant impact on HTM students’ CIs*.

#### 2.2.5. Research Setting

Different research contexts might result in slightly different outcomes. To eliminate these differences, we selected students’ grades as a characteristic of the study samples, before the COVID-19 outbreak or after, as time periods, and we treated them as moderator variables to test the differences.

Considering that the mastery of professional knowledge, the depth of exposure to professional work, the understanding of and expectations for jobs, and other relevant factors are different among students with different grades, it follows that there would be a discrepancy in the impact of different factors on their CIs. Many prior studies have examined students’ grades as a control variable [5,28]. Thus, to explore this difference, we took students’ grades as a source of variance to test the existence of differences in the factors’ effects on CIs. Given that students in their last two years of school have more opportunities for internship experiences and greater participation in the industry, the sample is divided into two categories: lower (freshman and sophomore) and higher (junior and senior). Thereby, the following hypothesis is proposed:

**Hypotheses** **14.***The influence of factors on HTM students’ CIs is different according to the students’ grades*.

As a typical service industry, the H&T industry has indeed been affected by the broad and profound adverse impacts of the COVID-19 pandemic [6,52], which is specifically reflected in the significant reductions in overall revenue, job shrinkage, reduced job stability, and so on. These adversities naturally have an impact on students’ perceptions and attitudes towards work in the H&T industry, causing them anxiety and a loss of confidence, thereby further reducing their commitment to the industry [6,7,19]. However, interestingly, there are also some studies have pointed out that HTM students still tend to choose jobs in the H&T industry during the pandemic, because they believe that the industry will eventually rebound [53]. So here we tested the moderating effect of the COVID-19 pandemic, and divided the sample studies into two categories according to the investigation time: before and after the outbreak of the pandemic.

**Hypotheses** **15.***The influence of factors on HTM students’ CIs is different according to the outbreak of COVID-19 pandemic*.

## 3. Methods

The meta-analysis method was first used in psychology in the 1970s to test proposed hypotheses [54], and it has now been widely introduced into the H&T field. Zhang et al. [55] used a meta-analysis approach to investigate the relationship between destination image and tourist loyalty. There were also interesting applications for exploring the enhancing effect of World Heritage on tourism development [56]. Meta-analysis methods have potential widespread applications in tourism research.

### 3.1. Data Collection

The meta-analysis data were collected from the empirical literature, including journal articles, book chapters, and conference papers, regarding the relationships between HTM students’ CIs and influencing factors by using the Web of Science, Google Scholar, EBSCO Hospitality & Tourism Complete database, Taylor & Francis database, and CNKI (a Chinese database). The last search was conducted in December 2021 and identified potential sources by using a combination of keywords, including the following: student or graduate; tourism or hospitality; career, employer/employment, or “job select/selection”; and intention. Meanwhile, we tried to collect as many of the references in the retrieved papers as possible. A total of 504 papers were retrieved. 

After the papers were collected, they were selected according to the following criteria: (1) empirical research; (2) conceptual clarity and reasonable construction of variables; (3) the subjects were identified as HTM students or graduates and the topic was CIs; and (4) the results reported the correlation coefficient or translatable indicators (e.g., standardized regression coefficient, path coefficient) between the influencing factors and CIs. The literature search resulted in a total of 34 papers comprising the meta-sample. Among them, only journal articles were included, as no other types of literature met the requirements; 30 were in English, and 4 were in Chinese (see Table 1).

### 3.2. Coding

For accuracy, the coding process was carried out in two steps. First, the descriptive statistics items and the effect–value statistics items of the literature were coded twice by one author at different times. The descriptive statistics included author, publication year, sample’s grades, independent variables (salary, social status, internship, etc.), dependent variable (CIs), and methods of data analysis (correlation analysis, regression analysis, structural equation analysis). The effect–value statistics included sample size, correlation coefficient, regression coefficient, and path coefficient. When no significant effect was reported, we coded the effect as insignificant or 0 throughout the database. Then, another author randomly selected 10 articles to recode and cross-check the process for consistency.

### 3.3. Statistical Analysis

The correlation coefficient was used to calculate the effect value of the meta-analysis. As some of the literature only reported the standard regression coefficient or path coefficient, we transformed them into the correlation coefficient by Peterson and Brown’s [70] method. Then, heterogeneity tests, publication bias analyses, effect–value combinations, and heterogeneity source analyses were performed on the extracted statistics. The above analysis process was implemented using Comprehensive Meta-Analysis software version 3.0.

## 4. Results

### 4.1. Publication Bias Analysis and Heterogeneity Test Results

Publication bias means that studies yielding results showing significant effects are more likely to be published in specific journals [71]. As the meta-analysis method is a comprehensive analysis based on the results of previous studies, the issue of publication bias should be considered. A fail-safe number is usually used to test whether publication bias exists, representing the number of additional unpublished studies that need to be found before the conclusion is overturned. As Rothstein et al. [71] suggested, a bias is present when a fail-safe number is less than *5k + 10 (k* is the number of studies). In this study, all the fail-safe numbers of the 13 independent variables were greater than the critical values (see Table 2), which indicated that no publication bias existed in the meta-analyzed samples.

We also tested heterogeneity, which was a prerequisite for the subsequent combined effects hypothesis test and for selecting the appropriate statistical model (fixed-effects and random-effects models). Here, we used the Q test and *I^2^* test, and a *p*-value of less than 0.05 and an *I^2^* value of more than 50% were required to be significant [72]. The heterogeneity of the effect values of all the influencing factors was significant (*p* < 0.05, *I^2^* > 50%) (see Table 2). As a result, the random-effects model could be used for the hypothesis testing.

### 4.2. The Test Results of Combined Effect

Table 3 shows the meta-analysis results of the relationship between each influencing factor and CIs. When the *p*-value was less than 0.05, the relationship was significant. Accordingly, each of the factors in the categories of work conditions and education have significant positive relationships with HTM students’ CIs in the H&T industry. This indicates that students generally take what jobs they can get in the H&T industry and what they perceive their education to be into consideration in their career choices. Therefore, H1, H2, H3, H4, H7 and H8 were supported. As for social identity and self-perception factors, both of them had a factor that was insignificantly associated with CIs, namely social status (*p* = 0.186) and outcome expectation (*p* = 0.052); a further consensus could not be reached. Thus, H6, H9, H10, H11 and H12 were supported, and H5 and H13 were not supported. Furthermore, according to Cohen’s [73] rule of judging the strength of a correlation, the correlation strengths, which were indicated by the combined effect size here, followed the order: career prospect (0.5 ≤ *r* ≤ 1, indicates a strong correlation); job satisfaction, person–organization fit, education quality, self-efficacy, environmental conditions (0.3 ≤ *r* < 0.5, indicates a moderate correlation); internships, salary, the nature of the work, and interpersonal relationships (0.1 ≤ *r* < 0.3, indicates a weak correlation).

### 4.3. Moderating Effect Test

As seen from Table 4, students’ grades explained the heterogeneity in the combined effect of the nature of the work, salary, environmental conditions, interpersonal relationships, career prospects, education quality, and internships (*p* < 0.05). Among them, in terms of the nature of the work, salary, interpersonal relationships and internships, the effect value of the students with higher grades is larger than that of the students with lower grades. Meanwhile, for environmental conditions, career prospects, and education quality, the situation is quite the opposite. H14 partially holds.

Since the collected sample studies after the beginning of the epidemic only addressed self-efficacy and education quality, only these two factors were tested here. Table 4 also shows that the relationships between the two factors and CIs were not significantly moderated by the COVID-19 pandemic; that is to say, no statistical changes in their influences on CIs were observed. The results were somewhat unexpected. Of note, even the influences of students’ self-efficacy were weakened after COVID-19.

## 5. Discussion and Conclusions

The viability and competitiveness of H&T enterprises are highly dependent on the stable workforce they employ [74]. One of the major aims for higher education institutions (HEIs) worldwide is to train talented professionals, playing a vital role in the H&T industries’ development [1]. Notwithstanding, the rapid growth of the H&T industries and the increasing scale of higher education in tourism and related majors do not go hand in hand. HTM students are less willing to be employed in it, thus leading to more brain-drain. This phenomenon has drawn the attention of scholars. Many studies have focused on students’ CIs and influencing factors, but their findings lack consistency and integration. After summarizing the selected 13 factors from the previous literature into four categories: work conditions, society identity, school education, and self-perception, we used a meta-analysis to integrate and compare these studies and quantitatively analyze the effects of influencing factors to seek systematic and integrated results. Furthermore, the relevant findings could provide implications for educators and industries to promote the intra-industry employment of HTM students.

The meta-analysis results showed that, in comparison, self-perception factors had a more prominent impact on HTM students’ CIs. Among them, job satisfaction had the largest positive effect, and it reflects the overall positive perception and attitude that an individual has toward his or her job. While job satisfaction and organizational commitment are two closely related concepts, high job satisfaction is usually accompanied by high organizational commitment. Furthermore, from the commitment theory perspective, people committed to an organization are willing to maintain a long-term relationship with the organization [75]. Therefore, when students are satisfied with their work during practical experiences such as internships, their intentions to stay in the H&T industry after graduation are stronger. Regarding person–organization fit, Song and Chathoth [25] identified it as a key factor influencing CIs. This study further confirmed this point; that is, students who are coadapted to the industry are willing to be employed within it. As Carless [76] noted, individuals are more successful and retain their chosen line of work when their personality traits and values match their jobs. A good person–organization fit could lead to many positive attitudes and behaviors, such as job satisfaction, organizational commitment, and better performance, which further encourage workers to stay in the industry or the organization [45,48]. Here, self-efficacy is also confirmed to be related to students’ CIs, which is consistent with previous studies [7,41,44]. According to social cognitive theory, when an individual believes in his or her abilities to accomplish a task, he or she will be motivated to do it [77]. Moreover, self-efficacy could be treated as an antecedent of job satisfaction [44]. Moreover, our research showed that employability is also significantly related to CIs. The cultivation of students’ employability is highly dependent on their education; however, there is a general disconnect between HTM education and the employability skills that the industries require [13,78]. Thus, developing students’ employability has become a common concern for communities, educational institutions, and industries [47]. Last, this study found no significant impact of outcome expectations. This is unsurprising, as previous studies have recorded both positive and negative impacts from outcome expectations [22,23], possibly partly attributable to the different backgrounds of the subjects. The development of the H&T industry in different countries or regions is different, and there is a big gap in the ability of industry jobs to meet job seekers’ expectations [64]. Moreover, students from different countries or regions have different employment concepts. In developing countries, jobs with a stable environment and a higher social status are more sought after by students [79]. Their job expectations are generally high, with jobs in the H&T industries considered difficult to achieve. However, students in developed countries are more likely to value friendly communication between colleagues and corporate culture [3]. In addition, the limited literature available for this meta-analysis may also be influential, and a consensus cannot be easily attained.

Correspondingly, HEIs plays an increasingly important role in forming students’ CIs and career decisions. Our results highlight the effects of education quality and internships. By directly affecting students’ mastery of professional knowledge, the development of employability and the formation of employment expectations further influence their willingness to stay in H&T [4]. Specifically, it is widely believed that higher education has a positive impact on students’ perceptions. The higher the perceived educational quality, the greater the academic self-efficacy of students, and the stronger their intentions to work in the industry [40]. Moreover, internships can provide students with a practical working scenario so that students can establish a clear understanding of the jobs. It confirmed that attitudes based on direct experience could better predict future behavior in H&T employment [15]. Notably, this study showed a weak correlation between internships and CIs, which is inconsistent with the increasingly important role of internship programs in students’ career development. The reason may lie in the different research contexts in the literature. Different cultural backgrounds might place differing emphasis on internships, while HEIs and enterprises have different arrangements for internships. 

The influence of social identity factors is partially confirmed. First, career prospects are closely related to the future development of students, and the identification of good prospects can help them build a higher industry commitment [5]. Nevertheless, HTM students and their relatives are generally not optimistic about the future of the H&T industry, resulting in a low willingness to work in it [1,15]. Hence, it is vital to shape a promising H&T industry in the minds of students and in society, which requires a joint effort between HEIs and industries. Meanwhile, we did not find a significant relationship between social status and CIs. This is possibly due to possible co-existing contradictions. One is that the social status of workers in the service industry, especially the H&T industry, is constantly improving with rapid development. With the government’s vigorous promotion and support of the H&T industry, people’s negative views of the industry are gradually disappearing; they believe work in this industry will be more respected and recognized in the near future [64]. The other is that people’s uncertainty and negative emotions about the H&T industries never dissipated, and in fact it became even worse after the pandemic’s outbreak [6,7]. In general, compared with self-perception factors, social identity factors have a weaker impact on students’ CIs, which is consistent with Armitage and Conner’s [80] views from the TPB. They pointed out that the subjective norm is generally found to be a weaker predictor of intention compared with the other two aspects.

Furthermore, the factor of work conditions was shown to be associated with CIs. The match between actual work conditions and students’ expectations of them directly affect their overall satisfaction. It further affects their willingness to work in the industry [15]. However, except for environmental conditions, the impacts of the other three factors are not as great as expected and are weakly related to CIs. Some scholars have also pointed out there is no significant relationship between salary and students’ CIs [5,14]. This indicates that HTM students now pay less attention to salaries when making employment decisions and that non-monetary incentives are more likely to help the H&T industry attract and retain HTM graduates. In addition, even though our study showed that interpersonal relationships have a relatively small impact on CIs, the H&T sectors should focus on fostering interpersonal identity, as this can satisfy one’s social exchange needs and makes one feel a sense of belonging [20]. Similarly, for the nature of the work, which is closely related to perceived social status and career prospects [5], students considered it as a key desirable attribute for jobs in the H&T industry [15]. Therefore, despite the lower impacts, it should also be given attention.

The heterogeneity of each factor was partially moderated. Students with higher grades place a greater emphasis on the nature of the work, salary, interpersonal relationships, and internships in their career decision-making process, as they have more access to work practices and practical experiences will deepen their exposure to industry work, making them focus on specific working conditions. More importantly, the higher a student’s grades, the deeper their understanding of the HTM program, which increases their involvement and positive views of H&T. Meanwhile, the enhancement of self-efficacy reinforces their self-judgement. In contrast, students with lower grades might think more about environmental conditions, career prospects, and education quality. That might be due to having little knowledge of the realities of industry work; they might obtain most of their information from parents, relatives, and teachers. The opinions of the people around them of their career prospects will greatly influence their career plans after graduation.

Interestingly, the COVID-19 pandemic did not impact the relationships between the influencing factors and CIs in this study. The reason for this may be partially due to having limited studies in the literature that met our requirements, and they merely discussed a few variables. Otherwise, the pandemic triggered a series of students’ negative emotions about the H&T industry, such as anxiety, tension, fear, etc., which were then reinforced by the situation. Additionally, the emergence of revenge tourism in the post-epidemic era, as well as the prevalence of rural and nature tourism and technology-based tourism [81,82], will somewhat dissipate students’ concerns about future industry development, so the impact of the epidemic on their CIs is correspondingly weakened. The differences in the factors’ impacts are not statistically significant. However, this is not to say that H&T educational institutions do not need to provide coping responses to the pandemic. We also identified that the relationship between students’ self-efficacy and their intentions to work in the industry has changed a little. More importantly, it has affected students’ expectations and judgments of future career development in the industry, resulting in low self-efficacy [7]. The importance of high-quality education needs to further highlighted in maintaining student hope and retaining students in the context of the epidemic [6,19].

## 6. Managerial Implication

Based on above conclusions, we propose some recommendations for HEIs and industries regarding the recruitment and maintenance of talent in the H&T industries. 

### 6.1. Implications for HEIs

The results show that students’ self-perception and school education are more influential in shaping students’ intentions to work, while school experiences play a large role in forming students’ career perceptions. Therefore, HEIs are instrumental in promoting students’ employment intentions, and should actively take measures to improve the quality of their H&T education. At first, some basic guarantees should be ensured, including recruiting and training highly qualified educators, improving educational infrastructures, and offering reasonable and effective curriculum to coordinate and balance the following three relationships: theory and practice; major and foundation; and knowledge and ability. Second, HEIs should place greater emphasis on students’ employability. Employability includes both soft and hard skills. Hard skills related to professional knowledge can be achieved through quality courses, while soft skills, including communication skills, social ability, creativity, etc., could be obtained by providing extracurricular activities (e.g., sports, music), creating a variety of clubs, and encouraging students to actively participate. In addition, with the prevalence of COVID-19 pandemic, HTM students’ concerns about careers in the H&T industry after graduation have increased significantly, which can lead to serious career anxiety. Thus, HEIs should offer psychological guides to appease and regulate students’ negative mindsets, and help them establish psychological coping mechanisms. HEIs should also improve the technical equipment for online teaching during the epidemic and train teachers to ensure proper teaching. A feedback mechanism should also be established to keep track of students’ satisfaction with the course and their level of knowledge. Adequate student–teacher interactions during quarantines are very important. Meanwhile, HEIs should increase lessons on business-related knowledge (e.g., marketing, consumer psychology), and cultivating students’ ability to unite and cooperate, which are more important in the labor market after the pandemic. To further alleviate HTM students’ concerns about job hunting after graduation and job uncertainty, HEIs can offer career guidance courses, expert consultation sessions, and innovative alumni associations to build career networks for current students.

### 6.2. Implications for Industry

Our study also demonstrated an association between the factors of working conditions and students’ CIs. H&T industries should consider their roles and responsibilities in promoting students’ employment intentions. Internships are a way for students to experience the working conditions of the industry first-hand. HEIs should design and implement reasonable and effective internship programs to clearly show students actual working conditions and the job content of H&T industries to help students establish realistic job expectations; moreover, H&T sectors should actively cooperate with HEIs to create positive work experiences for students and increase their satisfaction and loyalty. They should place a higher value on internship programs, and design content, sessions, processes, etc. efficiently, in order to improve students’ actual working capacity and to directly expand their identity and cognition about the H&T industries. Furthermore, students usually learn about actual working conditions from other practitioners, so the establishment of alumni associations could expose them to views of practitioners within the industry. Only staff who feel valued and respected, as well as receive favorable treatment from their job have a sense of commitment and positively influence others, improving the industries’ social acceptance and recognition and subsequently promoting students’ intentions to stay in the H&T industries. Therefore, H&T companies should tangibly improve conditions for their staff, such as salaries, incentives, working environments, communication channels, and promotion opportunities. In addition, in response to the shock of the COVID-19 pandemic, H&T companies should try their best to appease the anxiety and dissatisfaction of employees and improve their resilience by implementing recognition programs as well as providing training for crisis management skills and job rotation opportunities for their employees.

## 7. Limitations and Directions for Future Research

Some limitations of this study should be noted. First, given that the literature that strictly complied with the meta-analysis requirements and could be included was limited, the specific factors were tested by limited data. The number of sample studies should be expanded to further enhance the accuracy of the results in the future. Second, as subjected to the data and meta-analysis method, only four factor categories are addressed here. Other important and more specific factors should be explored, such as parental support, religions, etc. The education quality studied here is taken as a whole concept, which can be divided into separate aspects for research in the future, such as teachers’ methodology, curriculum setting, the provision of basic facilities, etc. At present, there are only a small number of quantitative studies about HTM students’ CIs during the COVID-19 pandemic, with a few specific influencing factors. In the future, we need to continue to obtain more sample studies to explore the impact of the pandemic. Lastly, students’ decisions about their major, such as deciding whether to take an internship or not, are important dynamics and need to be explored in the future. In addition, it is also worthwhile to study the moderating effect of the level of economic development by dividing research subjects into developed and developing countries.

## Figures and Tables

**Figure 1 behavsci-12-00517-f001:**
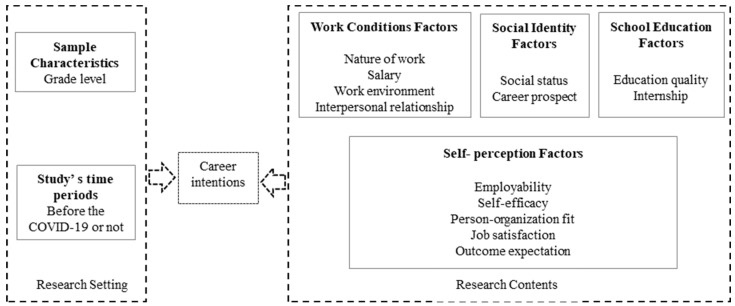
Proposed theoretical framework of career intentions and influencing factors.

**Table 1 behavsci-12-00517-t001:** Meta-analysis samples.

Author	Sample Size	Sample’s Grade	Influencing Factors
Kusluvan & Kusluvan (2000) [15]	397	Higher	nature of work(+); salary(+); environmental conditions(+); interpersonal relationship(+); social status(+); person-organization fit(+)
Wang et al. (2005) [23]	286	Higher	education quality(*); employability(+); outcome expectation(−)
Song et al. (2006) [57]	331	Higher	nature of work(+); salary(+); interpersonal relationship(*); job satisfaction(+)
Chuang et al. (2007) [58]	360	—	employability(+)
Teng (2008) [11]	483	Higher	nature of work(+); environmental conditions(+); interpersonal relationship(*); person-organization fit(+)
Song et al. (2010) [44]	352	Higher	interpersonal relationship(+); career prospect(+); self-efficacy(+); job satisfaction(+)
Chuang et al. (2010) [21]	131	Lower	self-efficacy(*); outcome expectation(+)
Wang (2011) [37]	686	Higher	career prospect(+); education quality(+)
Song et al. (2011) [25]	336	Higher	job satisfaction(+); person-organization fit(+)
Bai et al. (2012) [59]	720	Lower	social status(−); career prospect(+); education quality(+)
Chen et al. (2012) [60]	459	Higher	internship(+)
Lee et al. (2013) [20]	425	—	salary(+); interpersonal relationship(+)
Wan et al. (2014) [5]	205	—	nature of work(*);salary(*);social status(−);career prospect(+)
Seyitoğlu et al. (2015) [42]	305	Higher	education quality(+); internship(+)
Wang et al. (2014) [47]	245	Higher	internship(+); employability(+)
Walsh et al. (2015) [45]	246	—	job satisfaction(+)
Wang (2016) [22]	446	Lower	nature of work(+); social status(+); self-efficacy(*);outcome expectation(+)
Tan et al. (2016) [17]	70	Higher	nature of work(*); salary(+);social status(+)
Walsh (2016) [61]	387	Higher	salary(+); social status(+)
Park et al. (2017) [62]	307	—	salary(*)
Tsai et al. (2017) [41]	613	Higher	internship(*); self-efficacy(+)
Wen et al.(2018a) [28]	614	Lower	nature of work(*); salary(*); interpersonal relationship(*); social status(+); person-organization fit(+)
Wen et al. (2018b) [63]	525	Lower	internship(+)
El−Dief et al. (2019) [64]	227	—	environmental conditions(+); social status(*); internship(+); outcome expectation(+)
Boğan et al.(2020) [65]	474	Higher	person−organization fit(+)
Mensah et al. (2020) [66]	151	Higher	interpersonal relationship(+); internship(+)
Şengel et al. (2021) [26]	93	Lower	environmental conditions(+); person-organization fit(+)
Birtch et al. (2021) [7]	425	Lower	self-efficacy(+)
Kahraman et al. (2021) [40]	267	—	education quality(+)
Ghosh et al. (2021) [67]	140	Lower	internship(*)
Qu et al. (2021) [14]	207	Higher	salary(*); interpersonal relationship(+);internship(+)
Chen et al. (2021) [68]	918	Higher	employability(+)
Zhong et al. (2021) [19]	377	Higher	education quality(+)
Wang et al. (2021) [69]	782	—	internship(+); self-efficacy(+)

When more than 70% of the sample comprised freshmen or sophomores, the study was classified into the lower category, the higher category was labelled for juniors or seniors in the same way. “—”indicates that the sample proportions of the two categories were almost equal, or the samples cannot be identified. “+” = positive correlation; “−” = negative correlation; “*” = insignificant relationship.

**Table 2 behavsci-12-00517-t002:** Publication bias analysis and heterogeneity test results.

Categories	Factors	Number of Studies	Sample Size	Heterogeneity Test			*τ^2^*	Fail-Safe-Number
				Q Value	*p* Value	*I^2^*		
Work conditions	Nature of work	7	2546	115.544	0.000	94.807	0.053	148
	Salary	9	2943	163.123	0.000	95.096	0.061	227
	Environmental conditions	4	1200	14.237	0.003	78.927	0.014	114
	Interpersonal relationship	8	2960	62.289	0.000	88.762	0.022	75
Socialidentity	Social status	8	3066	310.706	0.000	97.747	0.119	138
	Career prospect	4	1963	152.549	0.000	98.033	0.110	706
School Education	Education quality	6	2641	243.215	0.000	97.944	0.113	898
	Internship	10	3654	239.188	0.000	96.237	0.073	744
Self-perception	Employability	4	1809	28.474	0.000	89.464	0.022	63
	Self-efficacy	6	2749	373.527	0.000	98.661	0.169	664
	Person-organization fit	6	2397	65.488	0.000	92.365	0.032	671
	Job satisfaction	4	1265	28.044	0.000	89.303	0.027	375
	Outcome expectation	4	1090	235.993	0.000	98.729	0.307	245

*I^2^* indicates the ratio of true heterogeneity to total variation in observed effects. *τ^2^* indicates the proportion of variation between studies that can be used to calculate the weight.

**Table 3 behavsci-12-00517-t003:** Results of random-effects model tests.

Hypothesis	Factors	Number of Studies	Sample Size	Combined Effect Size	Combined Effect Size Range 95%	Two-Tailed Test		Supported or Not
						*Z* Value	*p* Value	
H1	Nature of work	7	2546	0.176	0.002–0.340	1.979	0.048	Yes
H2	Salary	9	2943	0.184	0.018–0.339	2.174	0.030	Yes
H3	Environmental conditions	4	1200	0.321	0.197–0.434	4.919	0.000	Yes
H4	Interpersonal relationship	8	2960	0.128	0.019–0.234	2.292	0.022	Yes
H5	Social status	8	3066	0.163	−0.079–0.386	1.323	0.186	NO
H6	Career prospect	4	1963	0.516	0.238–0.716	3.407	0.001	Yes
H7	Education quality	6	2641	0.425	0.180–0.621	3.268	0.001	Yes
H8	Internship	10	3654	0.281	0.116–0.430	3.288	0.001	Yes
H9	Employability	4	1809	0.204	0.054–0.345	2.650	0.008	Yes
H10	Self-efficacy	6	2749	0.328	0.009–0.587	2.015	0.044	Yes
H11	Person-organization fit	6	2397	0.439	0.311–0.552	6.164	0.000	Yes
H12	Job satisfaction	4	1265	0.499	0.361–0.616	6.326	0.000	Yes
H13	Outcome expectation	4	1090	0.494	−0.801	1.943	0.052	NO

**Table 4 behavsci-12-00517-t004:** Results of random-effects model tests.

Factors	Grade Level					Research Time				
	Category	Number of Studies	*Q* Value	*p* Value	Combined Effect Size	Category	Number of Studies	*Q* Value	*p* Value	Combined Effect Size
Nature of work	Lower	2	4.065	0.044	0.042					
	Higher	4			0.286					
Salary	Lower	1	8.884	0.003	0.000					
	Higher	5			0.190					
Environmental conditions	Lower	1	10.738	0.001	0.575					
	Higher	2			0.284					
Interpersonal relationship	Lower	1	4.688	0.030	0.000					
	Higher	6			0.173					
Career prospect	Lower	1	35.445	0.000	0.760					
	Higher	2			0.446					
Education quality	Lower	1	11.154	0.001	0.710	Before	4	0.026	0.871	0.413
	Higher	4			0.368	After	2			0.450
Internship	Lower	2	4.082	0.043	0.086					
	Higher	6			0.301					
Self-efficacy	Lower					Before	5	2.221	0.136	0.371
	Higher					After	1			0.100
Person-organization fit	Lower	2	0.559	0.455	0.544					
	Higher	4			0.394					
Job satisfaction	Lower									
	Higher									

When examining the moderating effect of grades, only sample papers with clear classifications were included. Since the valid literature about some of the factors did not meet the standard of moderating effect test, they were not reported here.

## Data Availability

Data are available from the corresponding author upon reasonable request.

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
