# Peer review of "Factors Influencing Students’ Career Intentions in the Hospitality and Tourism Industries: A Meta-Analysis"

_behavsci, 2022, doi:10.3390/bs12120517_

Round 1
Reviewer 1 Report
This is a comprehensive study that has scoured the H&T literature for appropriate studies. The breadth and depth of their use of the literature is to be acknowledged. I would make two recommendations to strengthen the paper. First, discuss in some depth what's going on with the two hypotheses that were not supported (H2a and H4e). There is likely something interesting there and a rich discussion could shed some insights as to why no support was found. Second, I suggest that the authors add a new, clearly marked section (after the managerial implications section) that has a focus on curriculum development and/or H&T program implications. Sure, most of the data are about an individual's view of their work role; however, are there suggestions from the authors on what might H&T programs do to help the H&T industry retain employees longer?
Reviewer 2 Report
I am very honor to review this interesting and important study. Hospitality and tourism graduates’ vocational behavior and career development is vital to the sustainability of HTM education and relevant industries. This study focuses on the HTM students’ career intentions into hospitality and tourism industries and tries to find out the more important influencing factors by a meta-analysis. This is a valuable and interesting topic. The factors framework it constructed could reflect the structure of the influence factors and the existing literature. The meta-analysis is valid and appropriate way to test this hypothesis. In my opinion, it is significant contribution to improve the study of HTM students’ career behavior and publish here. However, I have some concerns.
1. Some literatures selected are talking about employment intentions. This study should explain the relationship with career intention more clearly.
2. The suggested "managerial implications" emerging from the paper are somewhat speculative and not directly supported by research or evidence.
3. The authors could discuss more the implications of the findings, especially the research implications or suggestions for future research.
4. In the manuscript, there are many abbreviations, such as H&T and HTM, when they are used first time, they must be explained.
5. Several references need to be carefully checked and corrected, please add some latest references.
Reviewer 3 Report
Summary
This study uses meta-analysis to get thorough findings on the quantitative links between the CI of HTM students and numerous influencing variables based on 34 selected journal articles. The authors categorize 11 influencing elements into 4 groups. Results highlight the psychological impact of the epidemic on students' career aspirations and intentions. Students' grades are also taken into account.
The article's primary contribution is that it seeks to advance existing studies by identifying the most crucial influencing factors, encouraging academics to come to an understanding, and inspiring them to develop more in-depth quantitative data analyses based on these 11 factors and other specific moderating variables like parental support and internship experience.
The paper is an excellent compilation of essential studies for researchers studying CI in connection with the H&T industry. Moreover, the article is a great summary of the current studies on the pandemic's modest impact on tourism students' CI.
Overall, the results of the meta-analysis combined with the given implications are contributing to H&T and behavioral sciences literature as well. The article has valuable recommendations for both HEIs and H&T companies.
Comments to the authors
There are only a few remarks that should be considered and revised before publication.
Table 1.: Having read the Kusluvan & Kuslvan article, it mentions person-industry congeniality as an important influencing factor, not person-organization fit, which is somewhat different. Therefore, it would be interesting to involve person & industry fit in the picture, as students often do not start with choosing an organization to work for. First, they have to decide whether it is a good choice in the long term to continue planning their career in the T&H industry after graduating. This could also affect how committed are they to the industry, another factor that is mentioned in the cited article. Environmental conditions are mentioned twice in this section, please revise the listed influencing factors again.
"nature of work(+); salary(+); environmental conditions(+); interpersonal relationship(+); environmental conditions(+); social status(+); person-organization fit(+)"
246-248: "However, interestingly, there are also some studies THAT have pointed out that HTM students still tend to choose jobs in the H&T industry during the pandemic, because they believe that the industry will eventually rebound." Does this result indicate that if we see the pandemic as a base for a crisis situation, the coming economic crisis might not cause different effects on CI either?
451-495: It was interesting to see results in light of the grades of the students. However, the impact of having working experience in the industry may show clearer results on CI, as many previous studies concluded. See Hb3.
Looking forward to reading the resubmitted article, good luck.
